# Increased contact transmission of contemporary Human H5N1 compared to Bovine and Mountain Lion H5N1 in a hamster model

Reshma Koolaparambil Mukesh[1] ✉, Franziska K. Kaiser [1], Jonathan E. Schulz[1], Shane Gallogly[1], Jessica Prado-Smith [2], Arthur Wickenhagen [1], Kathleen Cordova[2], Brian J. Smith[2], Chad Clancy [2], Carl Shaia [2], Greg Saturday [2], Emmie de Wit [1], Neeltje van Doremalen [1], Claude Kwe Yinda [1] & Vincent J. Munster [1]

The ongoing outbreak of highly pathogenic avian influenza virus (HPAIV) subtype H5N1 in the U.S. poses a significant public health threat. To date, 70 human cases have been confirmed in the United States, including two severe cases and one fatality. While suitable animal models are crucial for predicting the potential pandemic risk of newly emerging pathogens in humans, studies investigating contemporary HPAIV H5N1 transmission dynamics remain limited. Here, we investigate the pathogenicity and transmission efficiency of recent clade 2.3.4.4b H5N1 viruses isolated from a bovine, mountain lion, and a human case using Syrian hamsters. Intranasal inoculation results in productive virus replication in the respiratory tract and shedding for all three isolates. Transmission studies demonstrate limited efficiency via direct contact and airborne routes for all isolates. Although overall transmission is inefficient, the human H5N1 isolate demonstrates relatively greater contact transmissibility than the bovine and mountain lion isolates. Taken together, our findings demonstrate that the Syrian hamster model complements existing animal models for influenza A virus research and expands the resources available for investigating the pathogenicity, transmissibility, and efficacy of countermeasures against HPAIV H5N1.

Over the last decades, Highly Pathogenic Avian Influenza virus (HPAIV) H5N1 has evolved from a virus circulating mainly in poultry in Southeast Asia into a panzootic virus with circulation on all continents, including Antarctica[1–4]. The introduction of the clade 2.3.4.4b HPAIV H5N1 into the U.S. in 2021 was followed by rapid dissemination via migratory birds and spillover into a wide variety of mammals, backyard poultry and commercial poultry operations[4–6].

In the United States, until recently, the main mammalian species in which HPAIV H5N1 was detected were domestic and wild carnivorous species, including domestic cats (*Felis catus*), black bears (*Ursus americanus*), bobcats (*Lynx rufus*), red foxes (*Vulpes vulpes*), harbor seals (*Phoca vitulina*), and mountain lions (*Puma concolor*)[7]. In February 2024, the first detection of HPAIV H5N1 clade 2.3.4.4b, genotype B3.13 in dairy cattle was observed in Texas; from this initial

[1]Laboratory of Virology, National Institute of Allergy and Infectious Diseases, National Institutes of Health, Hamilton, MT, USA. [2]Rocky Mountain Veterinary Branch, National Institute of Allergy and Infectious Diseases, National Institutes of Health, Hamilton, MT, USA. ✉e-mail: reshma.mukesh@nih.gov

introduction, the virus has spread to 17 states and has affected over 1073 dairy herds (as of June 12, 2025)[8]. Within cattle, HPAIV H5N1 appears to be mainly replicating in the mammary tissue[9]. Transmission between dairy cattle is thought to occur via a combination of routes, including fomite transmission via contaminated milking equipment[10]. HPAIV H5N1 has been detected by PCR in milk within the food chain; however, several studies have confirmed the rapid inactivation of the virus by commonly used pasteurization methods[11–13]. In January 2025, a second independent introduction of HPAIV H5N1, clade 2.3.4.4b, genotype D1.1, in dairy cattle was identified in Nevada[14].

The significant increase of HPAIV H5N1 in the animals and environment resulted in spillover into humans. To date, 70 human cases have been reported, with most traced back to exposure to either dairy cattle or poultry[15]. In a few instances, the exposure history was unknown. HPAIV H5N1 in humans in the U.S. has generally caused mild illness, including conjunctivitis[16]. However, in 2025, the first fatal HPAIV H5N1 infection in the U.S. was reported, caused by a H5N1 clade 2.3.4.4b, genotype D1.1[17]. In several human cases, the HPAIV H5N1 exhibited markers of human adaptation including the E627K in PB2, and E186D and Q222H in the HA gene segments[18].

Currently, the clade 2.3.4.4b HPAI H5N1 viruses circulating in cows have retained their primary avian-type receptor preference[19], potentially due to the abundant expression of α2,3-linked sialic acid in the mammary glands of dairy cattle[20]. A limited number of amino acid substitutions in the H5 HA are believed to shift receptor binding preference from avian-type to human-type α2,6-linked sialic acids, a change associated with increased airborne transmission[21–23]. Despite this avian-type receptor preference of the HPAI H5N1 clade 2.3.4.4b viruses, efficient contact and airborne transmission have been observed in the ferret model, a model typically used to assess the human-to-human transmission efficiency of influenza A viruses[24–26]. To date, there is no evidence of human-to-human transmission associated with the ongoing outbreak in the United States.

Syrian hamsters have historically been developed as a small animal model for influenza A virus, and more recently as a model to study transmission[27–31]. H1N1 and H3N2 seasonal influenza A viruses are transmitted via the airborne route, as observed in the ferret model[32–34]. During the COVID-19 pandemic, the Syrian hamster model played a crucial role in understanding the drivers of human-to-human transmission of SARS-CoV-2[35,36]. In our study, we demonstrated the use of the Syrian hamster model to evaluate the transmission potential of three contemporary HPAIV H5N1 isolates obtained from a bovine, mountain lion and a human case: A/bovine/Ohio/B24OSU-342/2024, clade 2.3.4.4b, genotype B3.13 (Designated as Bovine), A/mountain lion/Montana/1/2024, clade 2.3.4.4b, genotype B3.6 (Designated as Mountain Lion) obtained from the lung tissue of a deceased mountain lion, and A/Texas/37/2024, clade 2.3.4.4b, genotype B3.13 (Designated as Human). We inoculated hamsters intranasally (I.N.) with $10^4$ TCID$_{50}$ of each of the three HPAIV H5N1 isolates and monitored disease progression and survival. Shedding kinetics and transmission efficiency were assessed for direct contact and airborne routes using virus detection and seroconversion. The Human isolate demonstrated productive transmission, with two of eight direct contact sentinels exhibiting active virus replication and shedding. In contrast, the Bovine and Mountain Lion isolates resulted in unproductive direct contact transmission in one of eight sentinels, with seroconversion but no virus detection. Unproductive airborne transmission was also observed in one of eight sentinels exposed to the Bovine isolate, resulting in seroconversion without virus detection. Our data demonstrates that, despite high levels of virus replication in the upper respiratory tract of hamsters, the efficiency of the Bovine, Mountain Lion and Human HPAIV H5N1 isolates to transmit via direct contact and airborne routes is limited.

## Results

### Productive infection of Syrian hamsters with contemporary HPAIV H5N1 isolates

To better understand the amino acid level differences among the three HPAIV H5N1 isolates (Bovine, Mountain Lion, and Human), we aligned the sequence of the respective gene segments. A summary of the amino acid variation among the three isolates is provided in Table S1. Among the three isolates, the PB2 E627K substitution was found exclusively in the Human isolate.

To evaluate the fitness of three contemporary HPAI H5N1 viruses in vivo, we inoculated Syrian hamsters ($n = 12$) I.N. with $10^4$ TCID$_{50}$ of each isolate and assessed replication and shedding kinetics (Fig. 1A). Six animals per group were scheduled for necropsy at four days post inoculation (dpi) to determine virus titers in the respiratory tissue samples. The remaining six animals per group were monitored daily for disease progression and survival.

Clinical signs of disease appeared from 2 dpi onward and included ruffled fur, hunched posture, hypoactivity, open-mouth breathing, dyspnea, and respiratory distress in all three groups. In the Bovine group, four animals began losing weight from 2 dpi. Of these, three reached predetermined euthanasia endpoint criteria on days five and six, while the fourth started to regain weight from day six onward (Fig. 1B). One animal from each of the Mountain Lion and Human groups succumbed to disease on day 2. Additionally, two animals from the Mountain Lion group and three from the Human group met euthanasia criteria between days four and seven (Fig. 1C, D). Overall, the survival after 21 dpi was 50% (3/6) in the Bovine (Fig. 1E) and Mountain Lion (Fig. 1F) groups and 33.33% (2/6) in the Human group (Fig. 1G). All surviving animals were seroconverted; however, endpoint antibody titers varied. The Bovine group had endpoint titers between 100 and 400, whereas the Mountain Lion group had titers ranging from 400 to 800. The two animals in the Human group had endpoint titers of 100 and 800 (Fig. 1H). Consistent with these results, neutralizing antibody titers were highest in the Mountain Lion group, whereas the Bovine and Human groups exhibited comparatively lower neutralizing titers (Fig. 1I).

### Hamsters inoculated with Human HPAIV H5N1 shed more infectious virus compared to Mountain Lion H5N1 isolate

Oropharyngeal and rectal swabs were collected daily to determine the kinetics of virus shedding. High viral RNA loads were detected by qRT-PCR in the oropharyngeal swab samples of all animals from all three groups with relatively similar shedding kinetics, viral RNA levels started to decline from 5 dpi onwards and no viral RNA was detected after 8 dpi (Fig. 2A). No significant differences were observed in total amounts of viral RNA shed as analyzed by area under the curve analysis (Fig. 2B). Corresponding to the amount of virus RNA, infectious virus was also found to be high during the earlier days of infection specifically on 1 and 2 dpi and started to decline thereafter (Fig. 2C). Significant differences were observed in the total amount of infectious virus shed, with the Human group shedding more infectious virus compared to the Mountain Lion group (Fig. 2D). Although viral RNA was detectable in rectal swabs, it was not consistently detected in all animals within each group (Fig. 2E). Analysis of the area under the curve revealed no significant differences in the total viral RNA shed from the intestinal tract among the three groups (Fig. 2F). In addition, infectious virus could only be found in one animal from the Bovine group (Fig. 2G), indicating that shedding from the intestinal tract is not prominent compared to shedding from the respiratory tract.

To determine virus replication in the upper and lower respiratory tract tissues, nasal turbinates (NT) and lungs were collected at 4 dpi. For six animals of each group, the relative lung/body weight was measured at necropsy on 4 dpi as an indicator of lung inflammation. In comparison with healthy control animals, a significant increase in relative lung weight was observed in the Human group (Fig. 2H). Virus

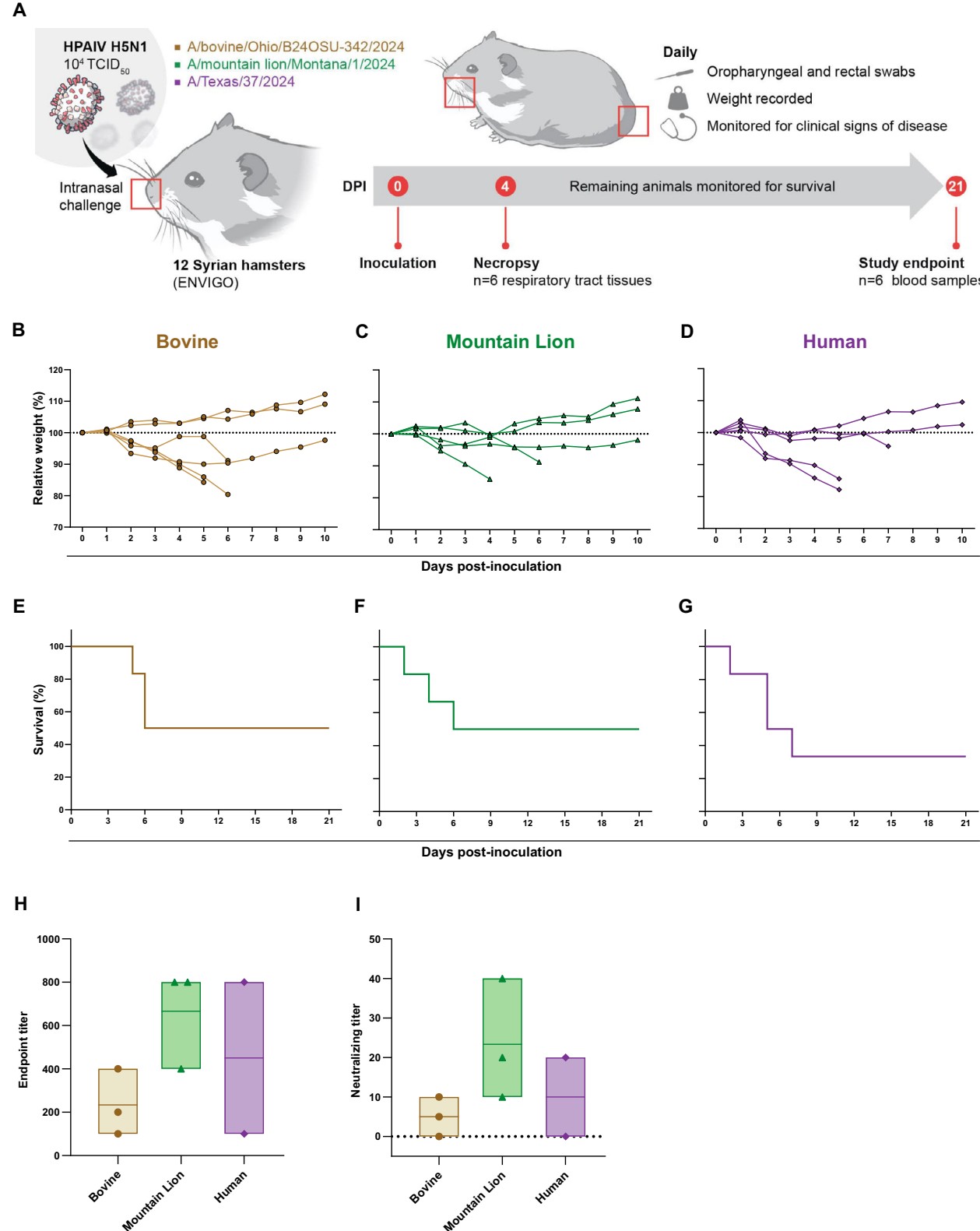

replication was detected in both the upper and lower respiratory tracts of all animals across the three groups. Although not significant, animals in the Bovine group had a relatively higher amount of viral RNA in the NT in comparison with Mountain Lion and Human groups (Fig. 2I). The amount of viral RNA in the lungs of infected animals was significantly higher in the Human group compared to the Mountain Lion group.

A similar pattern was observed for infectious virus levels in the lungs, with the Bovine and Human groups exhibiting, on average, higher viral loads compared to the Mountain Lion group (Fig. 2J). However, no significant differences in the amount of infectious virus in the NT and lungs were found between the three groups. Whereas high amounts of viral RNA were detected in the lung tissues of infected animals from all three groups, infectious virus could only be recovered

**Fig. 1 | Productive infection of Syrian hamsters with contemporary H5N1 influenza A viruses. A** Schematic overview of experimental design. Syrian hamsters ($n$ = 12 per group) were I.N. inoculated with $10^4$ TCID$_{50}$ of either A/bovine/Ohio/B24OSU-342/2024 (Bovine), A/mountain lion/Montana/1/2024 (Mountain Lion), or A/Texas/37/2024 (Human) in 40 μL of sterile DMEM. At 4 dpi, six animals from each group were euthanized for collection of respiratory tissues. The remaining six animals per group were monitored daily for clinical signs, weight loss, and survival up to 21 dpi. Oropharyngeal and rectal swabs were collected daily for 10 days. **B–D** Relative body weight changes over a 10-day period in hamsters infected with Bovine (**B**), Mountain Lion (**C**), or Human (**D**) HPAIV H5N1 isolates ($n$ = 6 per group). **E–G** Kaplan–Meier survival curves for each group. Survival proportions were compared using the Log-rank (Mantel-Cox) test. **H, I** Endpoint titers of humoral IgG anti-H5 HA responses in surviving animals (Bovine = 3, Mountain Lion = 3, and Human = 2) at 21 dpi by ELISA (**H**), and corresponding virus-neutralizing titers measured by microneutralization assay (**I**). For **H, I** data are represented as floating bars show the range from minimum to maximum, and the line within each bar denotes the mean. All individual data points are shown.

from a subset of these animals in the Bovine (3/6), Mountain Lion (1/6) and Human (4/6) groups. Overall, the Bovine and Human H5N1 viruses replicated to higher titers in the respiratory tract of the hamsters compared to the Mountain Lion H5N1 virus.

## Enhanced systemic spread of Bovine and Human H5N1 viruses compared to Mountain Lion H5N1

To assess systemic involvement after infection, six animals per group were inoculated with each isolate, and tissues were collected at 4 dpi. The spleen, liver, kidney, brain, and respiratory tract were harvested for subsequent virus titration and immunohistochemical analysis. Infectious virus was consistently recovered from the majority of respiratory tissues in all infected animals excluding lungs (Fig. 3A). Virus titers in the lungs differed significantly between the Mountain Lion and Human groups. While systemic dissemination was observed across all groups, infectious virus was not uniformly detected in extrapulmonary tissues from all animals. Notably, animals with lower viral titers in the respiratory tract also exhibited minimal or undetectable levels of infectious virus in peripheral organs (Table S2). Furthermore, the levels of infectious virus detected across all tissues were lower in animals from the Mountain Lion group than in those from the Bovine and Human groups. To further validate these findings, immunohistochemistry (IHC) was performed to detect the presence of Influenza A nucleoprotein on the same set of tissues.

The IHC results were consistent with the virological data, revealing tissue-specific pattern of virus antigen distribution (Fig. 3B). Consistent with infectious virus levels, variation in virus dissemination was observed among animals within the same group. Majority of animals in the Mountain Lion group showed no evidence of systemic spread, unlike those in the Bovine and Human groups. Virus antigen was detected in extrapulmonary tissues of 3 out of 6 animals in the Bovine group, 1 out of 6 animals in the Mountain Lion group, and 2 out of 6 animals in the Human group. Notably, systemic involvement was detected in animals with high viral antigen levels in respiratory tissues, as supported by virus titration data (Table S3). Overall, the Bovine and Human H5N1 viruses replicated to higher titers across most tissues compared to the Mountain Lion H5N1.

## More pronounced histopathologic lesions in the respiratory tract with Bovine and Human H5N1s compared to Mountain Lion isolate

While all three HPAIV H5N1 isolates replicated efficiently in the upper and lower respiratory tract of the hamsters, we were interested in whether the different viruses would display a differential disease phenotype and lung pathology. To better understand potential differences in disease phenotype, four animals per group were inoculated with $10^4$ TCID$_{50}$ of each virus and euthanized at 4 dpi. Respiratory tract tissues including NT, trachea, and lungs were collected for comprehensive histopathological evaluation of the entire respiratory tract. Histopathological evaluation revealed findings consistent with viral infection in hamsters (Table S4). The NT showed no inflammatory response in any of the groups, and minimal necrosis was observed in only two out of four animals from the Human group (Fig. 4A, B). Tracheal samples from all three groups contained varying amounts of degenerate or necrotic epithelial cells, along with abundant luminal

exudate composed of sloughed necrotic debris. Slightly more pronounced necrosis and neutrophilic tracheitis were observed in the Bovine and Human groups (Fig. 4A, C). Tracheal lesions were more pronounced in the Bovine ($n$ = 4/4) and Human groups ($n$ = 4/4). In contrast, the Mountain Lion group ($n$ = 2/4) had only two animals with minimal to moderate tracheal lesions (Fig. 4C). Pulmonary lesions were characterized by necrotizing broncho-interstitial pneumonia, consisting of multifocal inflammatory nodules centered on terminal bronchioles and extending into adjacent alveoli (Fig. 4A). The bronchiolar epithelium was frequently degenerate or necrotic, with abundant luminal exudate. These inflammatory nodules were composed predominantly of foamy macrophages, with fewer neutrophils and lymphocytes, and small amounts of necrotic debris. In most cases, hemorrhage, fibrin, and edema were present and intermingled with inflammatory cells, often extending into the surrounding alveoli. Adjacent alveoli were thickened due to the presence of fibrin, edema, and a small number of macrophages and neutrophils. Pulmonary pathology was more pronounced in the Bovine and Human groups. The Bovine group ($n$ = 3/4) had a mild to marked distribution of lesions while the Human group ($n$ = 2/4) had mild lesions (Fig. 4D). Pulmonary lesions were not observed in the Mountain Lion group. Overall, the Mountain Lion group exhibited comparatively mild histopathological lesions in the respiratory tract relative to the Bovine and Human groups.

To evaluate cellular tropism, IHC was performed to detect Influenza A nucleoprotein as a marker of virus replication. Viral antigen was present mostly on the respiratory and olfactory epithelial cells of the upper respiratory tract (Table S4). Cumulative IHC on the NT showed more staining in the Human group (3/4) compared to the Bovine and Mountain Lion groups (Fig. 4A, E). In the trachea, replication was mostly observed in the ciliated epithelial cells and cumulative IHC showed more staining in the Bovine and Human groups compared to the Mountain Lion group (Fig. 4A, E). Immunohistochemical evaluation of the lower respiratory tree revealed the presence of virus antigen in the bronchiolar epithelium, pneumocytes and alveolar macrophages (Table S4). Virus antigen could be detected in 4/4 hamsters in the Bovine, 2/4 hamsters in the Mountain Lion and 2/4 hamsters in the Human groups (Fig. 4A & E). Although the pathogenic phenotypes were overall comparable between the groups, slightly more influenza A virus replication was observed by IHC within the respiratory tract of hamsters in the Bovine and Human groups compared to the Mountain Lion group.

## Productive transmission of the Human HPAIV H5N1 via direct contact but not the airborne route

Recent analyses of the transmission kinetics of contemporary HPAI H5 viruses in ferrets have demonstrated increased viral shedding into the air, with efficient direct contact and partial airborne transmission[24,37]. However, within the current outbreak of Bovine HPAIV H5N1 in the U.S., no evidence of human-to-human transmission has been observed, and reported human cases have primarily resulted from exposure to infected livestock or poultry[16,38]. In this study, we aimed to investigate the transmission potential of three different HPAIV H5N1 isolates using the hamster model to complement existing data generated in the ferret model.

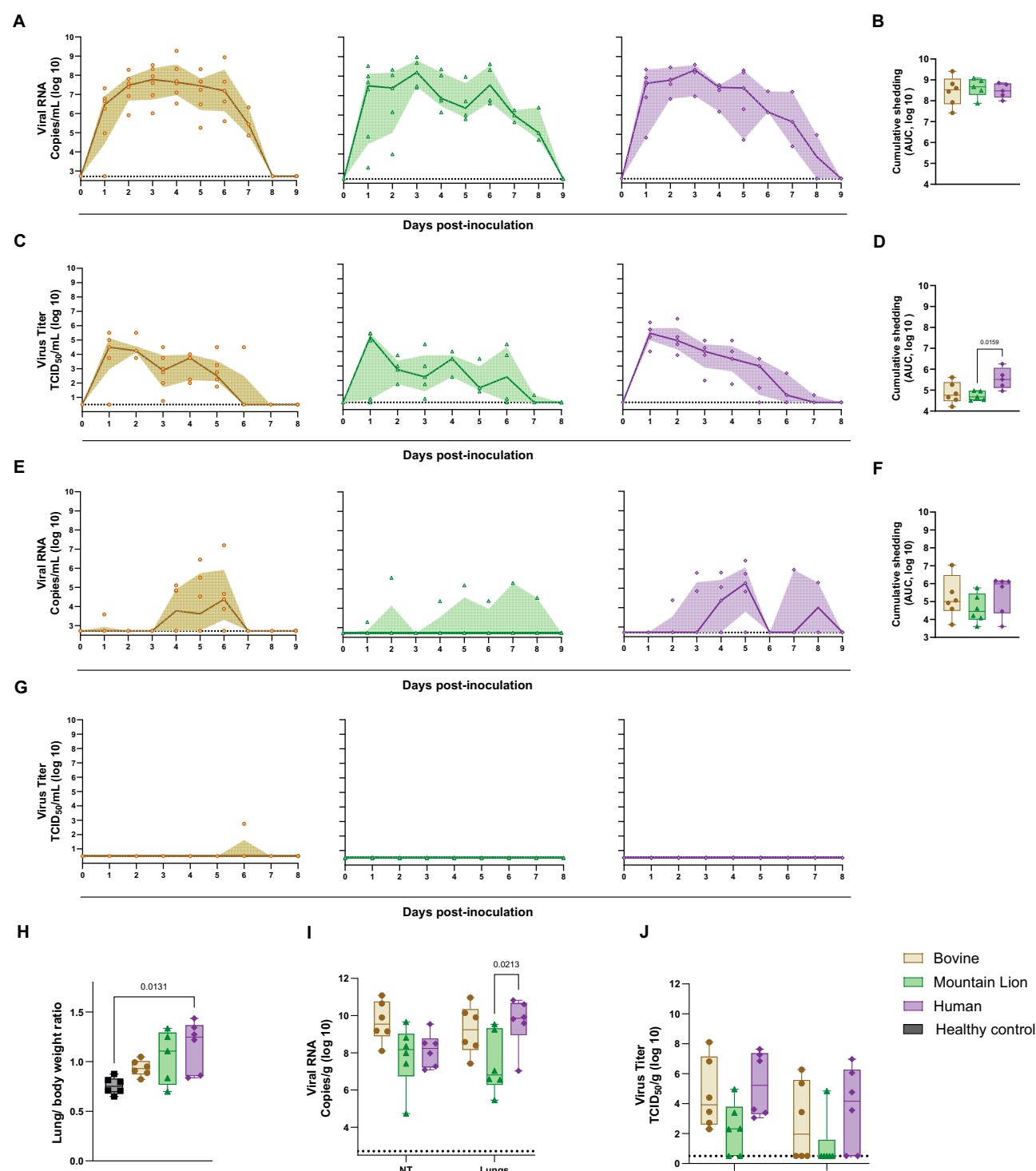

**Fig. 2 | Hamsters inoculated with Human HPAIV H5N1 isolate shed more infectious virus compared to Mountain Lion H5N1 isolate. A, B** Viral shedding kinetics as measured as RNA copy number in oropharyngeal swabs and corresponding area under the curve (AUC) analyses. Oropharyngeal swabs were collected daily, and viral RNA was quantified by RT-qPCR. The AUC of viral RNA shedding was calculated for each group. **C, D** Quantification of infectious HPAIV H5N1 in oropharyngeal swabs and corresponding AUC analysis. Infectious virus was measured by endpoint titration, and AUC was calculated to compare total oral shedding. **E, F** Viral RNA levels in rectal swabs and corresponding AUC analysis, as measured by RT-qPCR. **G** Infectious virus titers in rectal swabs. **H** Lung-to-body weight ratios at 4 dpi. Lungs were harvested from infected and uninfected control animals ($n = 6$ per group), and lung weight to body weight ratio was determined to

assess pulmonary pathology. **I, J** Nasal turbinates (NT) and lung tissues collected at 4 dpi were analyzed for viral RNA (**I**) and infectious virus (**J**). Statistical significance was assessed using a Kruskal-Wallis test followed by the Mann-Whitney U test for (**B**, **D** and **F**); Kruskal-Wallis test followed by Dunn's multiple comparisons test for H; and two-way ANOVA followed by Tukey's multiple comparisons test for I and J. $p$ value < 0.05, indicated where significant. Limit of detection was 2.72 $\log_{10}$ Copies/mL for (**A**, **E**, and **I**) 0.5 $\log_{10}$ TCID$_{50}$/mL for (**C**, **G**) and 0.5 $\log_{10}$ TCID$_{50}$/g for J. For (**A**, **C**, **E**, and **G**) individual values are shown; the line represents the median, and the shaded error bands indicate the interquartile range. For panels **B**, **D**, **F**, and **H–J** data are represented as box-and-whisker plots depicting the median (center line) and interquartile range (25th–75th percentiles). Whiskers indicate minimum and maximum values and all individual data points are shown. $n = 6$ per group.

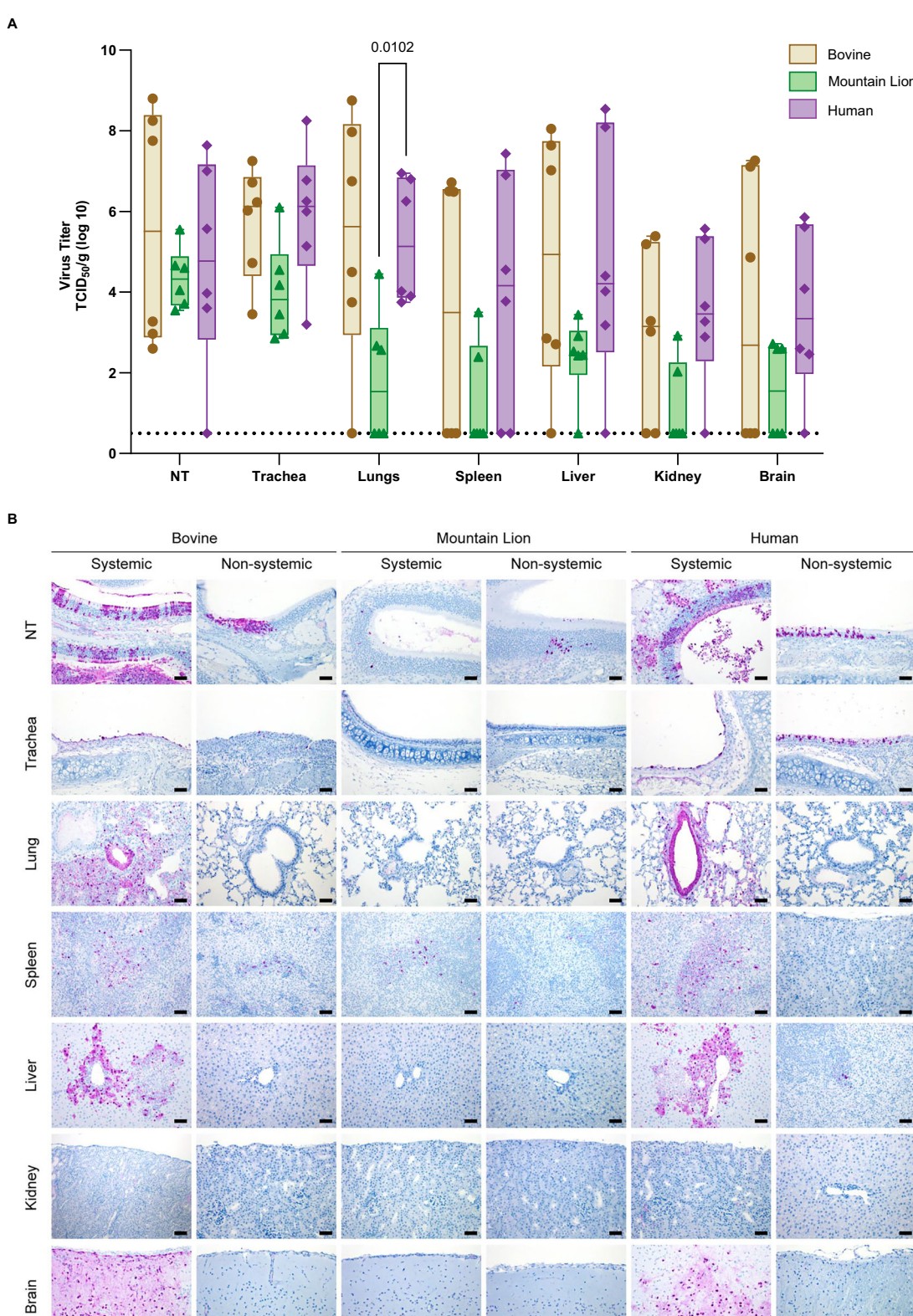

Eight donor animals per group were inoculated I.N. with $10^4$ $TCID_{50}$ of each virus. The transmission cage set-up used a cage divider, which allowed us to study direct-contact and airborne transmission within the same cage[36]. On day 0, inoculated donor animals were placed in the cage and 24 h thereafter naive direct-contact and airborne sentinel hamsters were introduced into the transmission cage. A total of eight individual transmission pairs were used to test the transmission efficiency. At the end of the 48-h transmission window (days 1 to 3 post-inoculation of the donors), the contact and airborne sentinel hamsters were single-housed, oropharyngeal swabs were collected daily and monitored for signs of disease. To assess transmission efficiency, detection of virus replication (viral RNA from the oropharyngeal swab samples) and seroconversion on day 21 after exposure (the presence of binding antibodies against H5) were used.

**Fig. 3 | Enhanced systemic replication of Bovine and Human H5N1 viruses compared to the Mountain Lion H5N1 virus. A** Hamsters (*n* = 6 per group) were inoculated with Bovine, Mountain Lion, or Human H5N1 viruses, and tissues were collected at 4 dpi. Virus titers were quantified and are presented as $\log_{10}$ transformed values of $TCID_{50}/g$. Statistical significance was assessed using Two-way ANOVA followed by Tukey's multiple comparisons test. *P* values adjusted for multiple comparisons and < 0.05 are indicated. Data are represented as box-and-whisker plots depicting the median (center line) and interquartile range (25th-75th percentiles). Whiskers indicate minimum and maximum values and all individual data points are shown. The limit of detection was $0.5 \log_{10} TCID_{50}/g$. Bars represent the range (minimum to maximum), with the middle line indicating the median. **B** Influenza A NP antigen distribution was assessed by IHC. Representative IHC images of Nasal turbinates (NT), trachea, lung, spleen, liver, kidney, and brain are shown from two animals per group, one exhibiting systemic involvement and one without. Magnification and scale bars: 200×, bar = 50 μm.

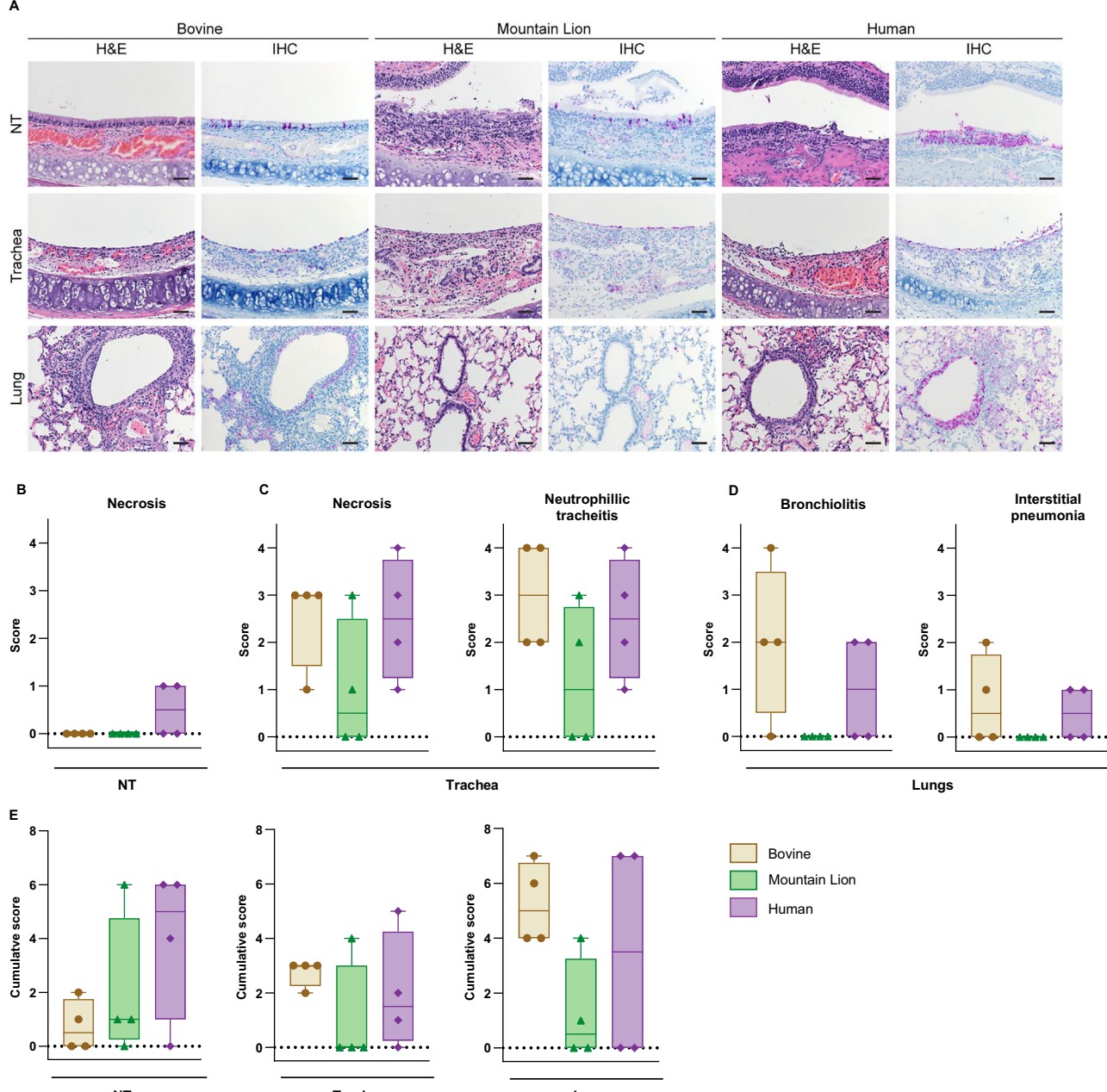

**Fig. 4 | More pronounced histopathologic lesions in the respiratory tract with Bovine and Human H5N1s compared to Mountain Lion isolate. A** Hamsters (*n* = 4 per group) were inoculated with Bovine, Mountain Lion, or Human H5N1 viruses, and tissues were collected at 4 dpi. Representative images of hematoxylin and eosin-stained (H&E) and immunohistochemistry (IHC) of Nasal turbinates (NT), trachea, and lung tissues at 4 dpi. Influenza A virus antigen was detected by IHC targeting influenza A virus nucleoprotein. Magnification and scale bars: NT: 400×, bar = 20 μm, Trachea: 200×, bar = 50 μm, and Lungs: 200×, bar = 50 μm.

**B–D** Semi quantitative pathological scoring of NT (B), trachea (C), and lungs (**D**). **E** Cumulative scoring of the presence of viral antigen in the respiratory tissues based on IHC staining. Statistical significance for panels (**B–E**) was assessed using the Kruskal-Wallis test. For graphs (**B–E**) data are represented as box-and-whisker plots depicting the median (center line) and interquartile range (25th–75th percentiles). Whiskers indicate minimum and maximum values and all individual data points are shown.

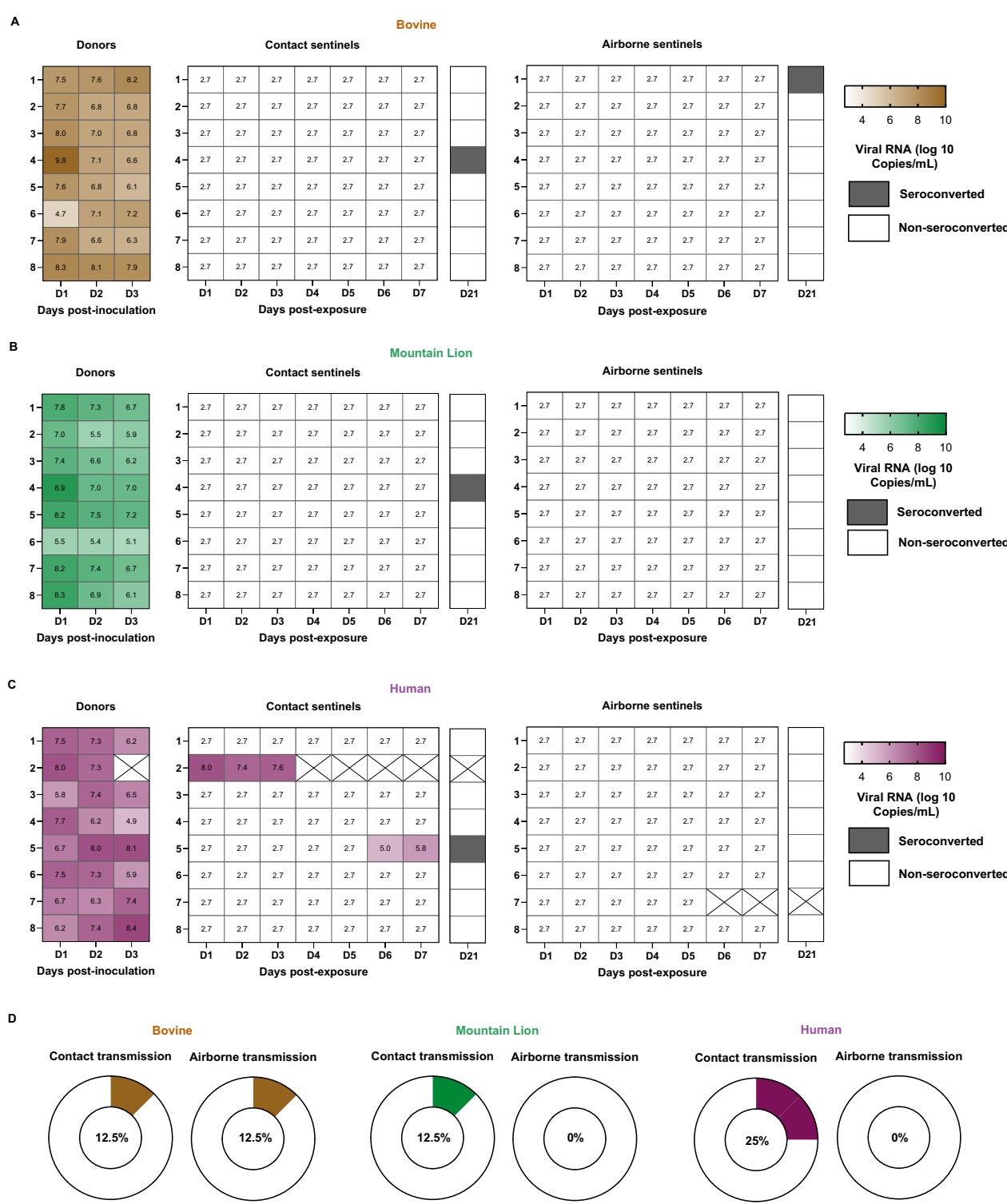

**Fig. 5 | Productive direct contact, but not airborne transmission of Human HPAIV H5N1. A–C** Shedding kinetics of viral RNA and seroconversion data obtained after transmission study. Donor hamsters ($n = 8$ per group) were intranasally inoculated with $10^4$ TCID$_{50}$ of one of three HPAIV H5N1 isolates: Bovine, Mountain Lion, or Human. At 1 dpi of donor animals, naïve sentinel animals were introduced into the transmission cages to assess virus transmission via either direct contact

($n = 8$ per transmission group) or airborne exposure ($n = 8$ per transmission group). Transmission was evaluated based on detection of viral RNA in oropharyngeal swabs and/or seroconversion at 21 dpi. **D** Summary of overall transmission efficiency. Transmission efficiency via direct contact and airborne routes for each virus is presented as donut plots. Limit of detection was 2.72 log$_{10}$ Copies/mL for A-C.

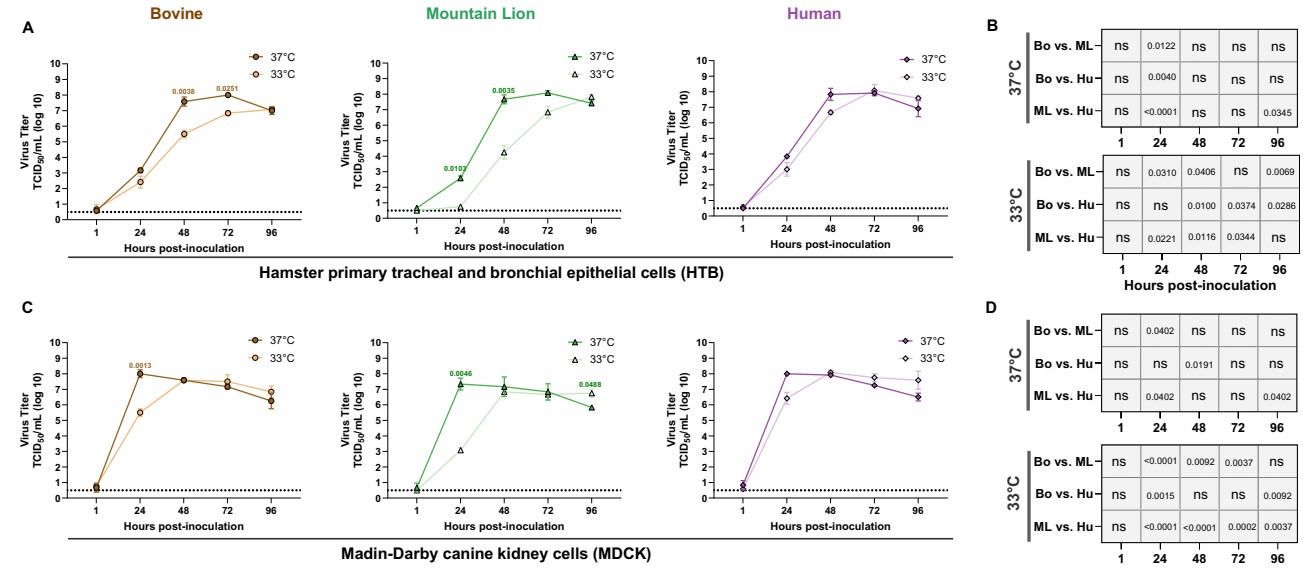

**Fig. 6 | Efficient replication of Human H5N1 compared to the Bovine and Mountain Lion H5N1 viruses.** A–D Growth kinetics of Bovine, Mountain Lion and Human H5N1 viruses in HTB (**A**, **B**) and MDCK cells (**C**, **D**) at 37 °C and 33 °C. Cells were infected at a multiplicity of infection (MOI) of 0.001 ($n = 3$ biological replicates), and supernatants were collected at different time points post-inoculation. Virus titers were determined and represented as $\log_{10}$ transformed values of TCID$_{50}$/mL. Statistical significance for panels A and C was assessed using the Two-way ANOVA with Sidak's multiple comparison test and B and D using Two-way ANOVA with Tukey's multiple comparisons test. Data are presented as mean with error bars indicating standard deviation. $p$ value < 0.05, indicated where significant. ns not significant. Bo Bovine; ML Mountain Lion; Hu Human.

All donor animals from the three groups were productively infected and shed viral RNA during the transmission window (Fig. 5A–C). Correspondingly, virus titers in oropharyngeal swab samples collected from donor animals during the transmission window were significantly higher for the Human isolate compared to the Mountain Lion isolate on day 2 (Fig. S1).

Despite the high levels of viral shedding from donor animals in all three groups, productive transmission via the direct contact route was observed only in the Human group (Fig. 5C). Two out of eight animals became productively infected, with high viral loads, similar to the donor animals, shed from the respiratory tract. Contact sentinel animal 2 in the Human group began showing signs of disease on 2 days post-exposure and reached euthanasia endpoint criteria on day 4. The remaining infected contact sentinel animal (Sentinel 5) in the Human group showed no severe signs of disease despite viral shedding and seroconverted. Full-genome sequence analyses of viruses from positive contact hamsters revealed no genetic changes between the original human isolate and the transmitted virus.

Contact sentinel 4 in both the Bovine and Mountain Lion groups seroconverted without showing any clinical signs of infection and no detection of viral RNA in oropharyngeal swabs. This suggests that virus transmission occurred but did not result in a productive infection, as it was quenched rapidly after transmission. Overall, the transmission efficiency via direct contact was 12.5% for the Bovine and Mountain Lion groups, and 25% for the Human isolate group (Fig. 5D).

Transmission via the airborne route was detected exclusively in the Bovine group (Fig. 5A). Sentinel animal 1 from the Bovine group seroconverted without detection of viral RNA in the oropharyngeal swabs, suggests that virus transmission did not result in a productive infection, but was quenched rapidly after transmission, comparable to the contact transmission of the Bovine group. Taken together, our results demonstrate that the Human HPAIV H5N1 has a higher transmission potential compared to the Bovine and Mountain Lion isolates, as productive transmission was observed only with the Human isolate.

## Human H5N1 replicates more efficiently than Bovine and Mountain Lion H5N1 viruses in hamster primary tracheal and bronchial epithelial cells (HTB) and Madin-Darby canine kidney cells (MDCK)

To assess the potential for increased transmission through enhanced virus replication, we conducted replication kinetics study using Human, Mountain Lion, and Bovine H5N1 viruses on HTB cells and MDCK cells at two different temperatures, 33 °C and 37 °C.

On HTB cells at 37 °C, Human isolate demonstrated efficient replication by 24 hpi, with significantly higher virus titers compared to the Bovine and Mountain Lion isolates (Fig. 6A, B). By 48 h, all three viruses reached similar titers with no significant differences. However, at 33 °C, a significant difference in virus titers between the Bovine and Human isolates was observed at 48, 72 and 96 hpi. Mountain Lion H5N1 showed enhanced temperature sensitivity, with reduced replication observed at 33 °C compared to 37 °C at both 24 and 48 hpi (Fig. 6A). Similarly, the Bovine H5N1 also exhibited temperature-dependent replication, with lower virus titers at 33 °C observed at 48 and 72 hpi.

A similar pattern of virus titers was visible in MDCK cells as well with Human H5N1 replicates faster than the Bovine and Mountain Lion H5N1s at both temperatures tested (Fig. 6C, D). Temperature sensitivity was evident with Bovine and Mountain Lion isolates. The Mountain Lion isolate showed reduced replication at 33 °C at 24 hpi, and the Bovine isolate demonstrated a similar trend at this time point (Fig. 6C). In contrast, the Human H5N1 isolate replicated efficiently at both 33 °C and 37 °C across all time points tested, with no significant differences in viral titers, indicating minimal temperature sensitivity.

In conclusion, among the three isolates, the Human H5N1 virus exhibited the most efficient replication and the least sensitive to temperature variation, highlighting its enhanced replication fitness compared to the Bovine and Mountain Lion isolates.

## Discussion

The zoonotic and human-to-human transmission potential of HPAIV H5Nx viruses has been under investigation since the first human cases

were reported in 1997 in Hong Kong[39]. The introduction of HPAIV H5N1 in the Americas and subsequent outbreaks in poultry and dairy cattle have dramatically increased human exposure to these viruses[40,41]. The majority of human cases in the U.S. stem either from exposure through infected poultry or dairy cattle. However, sustained human-to-human transmission of HPAIV H5N1 has so far not been reported[42].

Since the first detection of the HPAIV H5N1 clade 2.3.4.4b in dairy cattle, several studies have investigated the pathogenicity and transmission potential of HPAIV H5N1 isolates using a variety of animal models including mice, ferrets and non-human primates. Among the variety of mouse models utilized, the HPAIV H5N1 typically causes rapid generalized disease with systemic replication and uniformly lethal outcomes, regardless of the mouse strain or exposure route[24,43,44]. In addition, $10^6$ plaque-forming units (PFU) of the Bovine and Human H5N1 isolates caused severe disease in ferrets resulting in rapid disease progression, systemic replication, and uniform lethality[24,25]. The pathogenicity associated with $10^7$ TCID$_{50}$ of the Bovine H5N1 isolate in non-human primates was dependent on the inoculation route. While intratracheal inoculation resulted in severe, fatal broncho-interstitial pneumonia, intranasal or orogastric exposure led to mild or subclinical disease. Regardless of the route of inoculation, all animals shed virus both orally and nasally[45].

While hamsters are a well-established small animal model for influenza A virus research, they are not as widely used as ferrets or Guinea pigs[27,46]. In comparison to the ferret model, hamsters are easier to handle, cost-efficient, and require less housing space. Although guinea pigs are considered an alternative animal model, they typically show less clinical signs or pathogenicity after infection with pathogenic viruses such as HPAI H5N1[47,48]. Moreover, Syrian hamsters have been extensively used to evaluate the transmission of several respiratory pathogens, including SARS-CoV-2 and Nipah virus, making them an attractive animal model for investigating transmission[35,36,49]. A recent hamster study evaluated the pathogenicity and transmission potential of the Human H5N1 isolate using $10^3$ PFU in 30 µL of PBS, and demonstrated that hamsters are susceptible to HPAIV H5N1, showing high lethality and 100% transmission efficiency via direct contact[50].

In our study, intranasal inoculation with $10^4$ TCID$_{50}$ resulted in 50% survival in both the Bovine and Mountain Lion groups and 33.3% survival in the Human group. Infected hamsters displayed differences in shedding and replication between the three isolates. Interestingly, cumulative respiratory shedding of viral RNA did not differ significantly among the three groups. However, higher levels of infectious virus were detected in the Human group compared to the Mountain Lion group, suggesting potential differences in viral replication efficiency or host-pathogen interactions despite similar RNA shedding profiles. These findings are consistent with the observed trends in viral RNA levels in the URT, where the Bovine group exhibited higher RNA loads, while comparable levels were detected between the Human and Mountain Lion groups. Notably, despite similar RNA loads, differences in infectious virus titers in the oropharyngeal swab samples were observed only between the Human and Mountain Lion groups, indicating potential differences in viral replication competence or shedding of infectious particles. Although these viruses replicated efficiently and were shed from the respiratory tract of hamsters, their relative transmission efficiency remained low. This discrepancy may be attributed to the inefficient release of infectious viral particles into the air, limiting onward transmission. Among the three isolates tested, productive transmission was observed only with the Human isolate. This observation is supported by a recent study in ferrets which displayed a higher release of infectious virus in the air with the Human HPAIV H5N1 than Bovine HPAIV H5N1[25,51].

The Human isolate, A/Texas/37/2024, has now been utilized in a wide variety of in vitro and in vivo studies, resulting in a better understanding of the phenotypic characteristics of this virus. The A/Texas/37/2024 isolate contains the PB2-E627K substitution, a mutation known to enhance the replication capacity of influenza viruses in mammalian hosts[52]. In A/Texas/37/2024, the PB2-E627K increases the viral polymerase activity in minigenome assays[37]. The PB2-E627K substitution is a common occurrence and has been identified in multiple human HPAIV H5N1 cases, independent of exposure source (e.g., dairy cattle or poultry), or lineage (Clade 1, Clade 2.2, and Clade 2.3.4.4b)[53,54]. In addition, recent reverse-genetics study confirmed the PB2-E627K substitution in the human isolate as a key determinant in enhancing viral polymerase activity, resulting in increased replication and pathogenicity in C57BL/6 J mice[52]. In our study, the Bovine and Human isolates exhibited a more pathogenic phenotype compared to the Mountain Lion isolate. Although all three viruses replicated throughout the upper and lower respiratory tract, the Bovine and Human isolates induced more extensive pulmonary lesions compared to the Mountain Lion isolate. This difference may be attributable to the distinct genetic makeup of the Mountain Lion isolate (genotype B3.6).

Historic and contemporary HPAIV H5N1 isolates have been extensively characterized in the ferret model for their potential to transmit through direct contact and airborne routes. In the ferret model, the Bovine isolate did not transmit efficiently via the airborne route, whereas the Human isolate transmitted efficiently via direct contact and airborne transmission routes in two separate studies[24,25]. A similar study evaluating the transmission efficiency of an H5N1 virus isolated from a mink in Spain, demonstrated efficient transmission potential via direct contact (75%) and air (37.5%) in ferrets[26]. Typically, transmission events resulted in fatal disease in recipient ferrets. Collectively, these studies demonstrate relatively high transmission efficiencies of HPAIV H5N1 (37.5%–100%); however, it is unclear how this correlates with the potential for human to human transmission. Within our study, only direct contact transmission with the Human isolate resulted in productive infection and seroconversion. In contrast, one direct contact sentinel each from the Bovine and Mountain Lion groups seroconverted without detectable replicating virus in oropharyngeal swabs. For the airborne route, only a single sentinel in the Bovine group was seroconverted, again without the detection of replicating virus in the oropharyngeal swabs. Our results are in agreement with a recent study that only used the Human isolate and displayed contact but not airborne transmission in the hamsters, albeit at a higher efficiency than in our study[50].

Experimental modelling of transmission is not standardized across laboratories, and variation in the transmission setup, exposure duration, animal species, temperature and humidity could affect the efficiency of transmission. Our study conducted with the Syrian hamster model employed validated transmission caging and utilized ALPHA-dri bedding to minimize fomite transmission while under directional airflow conditions. This approach might be different from other hamster or ferret transmission experiments. In particular, housing, airflow, inoculum volume and dose, and host variables such as age and sex could result in variability between studies. These methodological and biological differences underscore the importance of context when comparing transmission efficiency across different studies and animal models.

The relatively inefficient transmission in our study aligns with the lack of reported human-to-human transmission. Comparable to other studies, increased viral shedding with the Human isolate is associated with a higher transmission potential. In our study, among the three H5N1 isolates tested, the Human H5N1 demonstrated the most efficient replication in vitro and in vivo, along with minimal sensitivity to temperature differences. In contrast, the Bovine H5N1 showed moderate replication efficiency, while the Mountain Lion H5N1 was the least efficient and exhibited limited systemic spread in vivo. Additional factors may influence transmission efficiency, including host immune responses such as interferon antagonism or cytokine modulation.

Despite the productive infection in the upper respiratory tract of infected hamsters, the relatively low transmission efficiency of these

isolates suggests that additional phenotypic changes are necessary for increased transmission. The relative receptor distribution pattern in hamsters is comparable to that in other species, including humans, with the expression of both α2,6- and α2,3-linked sialic acid receptors in the respiratory tract, albeit at different ratios[27]. Recent studies have demonstrated that the Bovine and Human isolates maintain avian-type receptor-binding specificity[55]. In addition, in vitro studies have shown that the Q226L and N224K substitutions in the HA of previous human HPAIV H5N1 isolates shift receptor specificity toward α2,6-linked sialic acids, potentially enhancing the transmission potential. This observation aligns with findings from the ferret model, where alterations in receptor specificity have been linked to efficient airborne transmission of HPAIV H5N1[21–23].

The ongoing circulation of HPAIV H5 in wild birds, wild mammals, poultry and dairy cattle is cause for considerable concern. The continued circulation of these viruses significantly heightens the risk of adaptations that could enhance zoonotic potential and facilitate more efficient human-to-human transmission. A combination of in silico, in vitro, and in vivo approaches is needed to assess the enhanced zoonotic potential of newly emerging H5 variants. Similar to the SAVE program established for SARS-CoV-2, real-time risk assessment of emerging mutations in HPAIV H5N1 is essential to evaluate their potential impact on pathogenicity, transmissibility, and the effectiveness of available countermeasures[56]. The hamster model complements established animal models, such as the ferret model, and broadens the available tools for studying the pathogenicity and transmission dynamics associated with HPAIV H5N1 infection.

## Methods

### Biosafety and biocontainment statement

This study was reviewed and approved by the NIH RML Institutional Biosafety Committee (IBC) and the NIH Dual Use Research of Concern-Institutional Review Entity (DURC-IRE). The project did not meet the US Government definition of DURC or ePPP. All work with infectious HPAI H5N1 viruses was approved for biosafety level 3 (BSL-3) conditions by the IBC. This study involved experimental infection of naïve hosts (Syrian hamsters) with HPAI H5N1 to address critical questions related to virus pathogenesis, host response, and transmission-relevant outcomes. Our experimental work will help to better understand why these viruses can replicate in these hosts to develop appropriate countermeasures against these viruses. The scientific benefits of this work include improving understanding of pathogenicity and transmission that are directly relevant to public health preparedness, risk assessment, and the development of effective countermeasures. Given the inherent risks associated with working with HPAIV, all experiments were conducted under appropriate biocontainment conditions (BSL3) in accordance with institutional biosafety regulations. All sample inactivation procedures were performed by trained and authorized personnel in accordance with IBC-approved standard operating procedures for the removal of specimens from high-containment facilities.

### Virus and cells

Human HPAIV H5N1 isolate A/Texas/37/2024 (EPI_ISL_19027114) was obtained from Dr. Todd Davis at the Centers for Disease Control and Prevention, Decatur, Georgia, USA. Bovine HPAIV H5N1 isolate A/bovine/Ohio/B24OSU-342/2024 (EPI_ISL_19178076) was obtained from Richard Webby at St. Jude's Children hospital, Memphis, TN, USA and Andrew Bowman at Ohio State University, Columbus, OH, USA. Mountain lion HPAIV H5N1 isolate A/mountain lion/Montana/1/2024 (EPI_ISL_19083124) was obtained from a lung sample of a diseased mountain lion in Montana in February 2024. All three viruses belong to the 2.3.4.4b clade of HPAIV H5N1. The Bovine and Human isolates are classified as genotype B3.13, whereas the Mountain Lion isolate belongs to genotype B3.6.

Virus propagation was performed in MDCK cells in MEM supplemented with 1 mM L-glutamine, 50 U/mL penicillin, 50 µg/mL streptomycin, 20 mM HEPES, and 4 µg/mL TPCK trypsin. MDCK cells were maintained in MEM supplemented with 10% fetal bovine serum, 1 mM L-glutamine, 50 U/mL penicillin, and 50 µg/mL streptomycin, 20 mM HEPES. Mycoplasma testing is performed at regular intervals. No mycoplasma was detected during the study.

Replication kinetics study was performed on HTB and MDCK cells. HTB cells (Cat# HM-6033) were obtained from Cell Biologics and maintained according to the manufacturer's instructions.

### Animal studies

Twelve male and female Syrian hamsters (4-6 weeks old; ENVIGO) were intranasally inoculated with either the HPAIV H5N1 Bovine, Mountain Lion, or Human isolate. I.N. inoculation was performed with $10^4$ $TCID_{50}$ of each isolate in 40 µL sterile DMEM. Post-inoculation, all animals were weighed daily and monitored for clinical signs of disease. Oropharyngeal and rectal swabs were collected daily to assess the virus shedding dynamics. 4 dpi, six hamsters from each group were euthanized and respiratory tract tissues were collected. Six remaining animals were monitored for survival until 21 days, and blood samples were collected after euthanasia.

To address the histopathological changes associated with Bovine, Mountain Lion, or Human isolate, four male and female Syrian hamsters (4–6 weeks old; ENVIGO) were inoculated I.N with $10^4$ $TCID_{50}$ of each H5N1 isolates in 40 µL sterile DMEM. 4 dpi, respiratory tissues including NT, trachea and lungs were harvested. H&E staining and IHC analysis were performed to understand the histopathological changes and the virus distribution.

### Transmission study

Specially designed cages were used to perform transmission studies. Transmission cages were divided by 3D-printed perforated plastic dividers to allow airflow from the inoculated to the naïve hamster but prevent direct contact and fomite transmission[36]. Donor hamsters ($n = 8$) were inoculated I.N. as described above with $10^4$ $TCID_{50}$ of each HPAIV H5N1 isolate. On day 1 post inoculation, donor animals were placed on one side of the cage together with the naïve contact sentinels ($n = 8$). Likewise, one naïve animal was placed on the other side of the cage and served as the airborne sentinel ($n = 8$). The divider placed between the cage allowed continuous airflow from the infected animal to the naïve airborne sentinel. Hamsters were cohoused for 48 h. The following day (D3), donor animals were euthanized, and sentinel animals were rehoused into regular rodent cages. Oropharyngeal swabs were collected for eight days. On day 21, animals were euthanized, and blood samples were collected.

### Viral RNA detection by RT-qPCR

Viral RNA was detected by qRT-PCR. RNA was extracted from swabs using a QiaAmp Viral RNA kit (Qiagen) according to the manufacturer's instructions. Lung and NT tissues were homogenized, and RNA was extracted using the RNeasy kit (Qiagen). Viral M gene-specific primers, Forward primer: AAGACCAATCCTGTCACCTCTGA, Reverse primer: CAAAGCGTCTACGCTGCAGTCC and Probe: FAM-TTTGTGTTCACGCTCACCGTGCC-TAMRA (Integrated DNA Technologies) were used for the detection of viral RNA. RNA (5 µl) was tested using the TaqMan Fast Virus One-Step Master Mix (Applied Biosystems) and the QuantStudio 3 Flex Real-Time PCR System (Applied Biosystems) according to the manufacturer's instructions. Dilutions of Influenza standards with known genome copies were run in parallel to prepare a standard curve and calculate copy numbers/mL or copy numbers/g. The detection limit for the assay was 5 copies/reaction, and samples below this limit were considered negative.

## Virus titration

Infectious virus present in the swab and tissue samples was quantified by endpoint titration of 10-fold dilutions on MDCK cells in 96-well plates. Tissue samples were homogenized in 1 mL media using TissueLyser II (Qiagen). Cells were washed twice prior addition of 10-fold serially diluted swab samples and incubated the cells for 3 days at 37 °C and 5% $CO_2$. Three days post-incubation, the presence or absence of infectious virus was measured by a standard HA assay using turkey red blood cells (Innovative Research). RBCs were washed at least three times with PBS and diluted to 0.33% in PBS directly before use. 75 μL of 0.33% RBCs was added to 25 μL of virus and incubated for 1 h at 4 °C. Subsequently, wells were marked as either agglutinated or negative. Titers were calculated using the Spearman-Karber Method.

## Enzyme-linked immunosorbent assay

Nunc MaxiSorp flat bottom 96-well plates (ThermoFisher Scientific) were coated with 50 ng in 50 μL/well of influenza A H5N1 A/Vietnam/1203/2004 hemagglutinin (HA) protein (IBT Bioservices) and incubated overnight at 4 °C. Next day, the plates were blocked with 100 μL of casein in PBS (ThermoFisher Scientific) for 1 h and incubated with serially diluted hamster sera (1:100-1:512000, in duplicate) for 1.5 h at room temperature. Immunoglobulin G (IgG) antibodies were detected by using affinity-purified polyclonal antibody peroxidase-labelled anti-hamster IgG (Seracare CAT# 5220-0371) at a dilution of 1:2500 in casein followed by 3,3′,5,5′- Tetramethylbenzidine 2-component peroxidase substrate (Seracare, 5120–0047) and stop solution (Seracare, 5150-0021). The optical density at 450 nm (OD450) was measured. Serological analysis of samples from the transmission study was performed at a 1:100 dilution. The threshold for positivity was determined as the average plus three times the standard deviation of the negative control hamster sera.

## Virus neutralizing antibody assay

Irradiated serum samples were heat-inactivated, two-fold serially diluted, and incubated with 100 $TCID_{50}$ of corresponding influenza viruses for 1 h at 37 °C. The virus-serum mixtures were added to confluent MDCK cells in 96-well plates and incubated at 37 °C and 5% $CO_2$. Three days post-incubation, the presence or absence of infectious virus was measured by a standard HA assay using fresh turkey red blood cells (Innovative Research). Neutralization titers were defined as the reciprocal of the highest serum dilution at which hemagglutination was absent.

## Histology and Immunohistochemistry

Tissues were fixed in 10% neutral buffered formalin x2 changes for a minimum of 7 days. Tissues were placed in cassettes and processed with a Sakura VIP-6 Tissue Tek on a 12-h automated schedule, with a graded series of ethanol, xylene, and PureAffin. Embedded tissues were sectioned at 5 um and dried overnight at 42 °C before staining. Immunoreactivity was detected using Millipore Sigma Anti-Influenza A nucleoprotein antibody at a 1:12,000 dilution. Roche Tissue Diagnostics DISCOVERY Omnimap anti-rabbit HRP was used as a secondary antibody. For negative controls, replicate sections from each control block were stained in parallel following an identical protocol, with the primary antibody replaced by Vector Laboratories rabbit IgG at a 1:2500 dilution. The tissues were stained using the DISCOVERY ULTRA automated stainer (Ventana Medical Systems) with a Roche Tissue Diagnostics DISCOVERY purple kit.

## Replication kinetics

To evaluate replication kinetics, HTB and MDCK cells were infected with Bovine, Mountain Lion, or Human HPAI H5N1 viruses at a multiplicity of infection (MOI) of 0.001. Infections were carried out for 1 h at either 37 °C or 33 °C with 5% $CO_2$, with gentle rocking every 15 min. Following infection, inoculum was removed, cells were washed three times with 1 mL of DPBS, and fresh culture medium was added. Supernatants were collected at 1, 24, 48, 72, and 96 h post-infection and virus titers were determined using $TCID_{50}$ assay on MDCK cells.

## cDNA synthesis

Virus RNA was reverse transcribed using the SuperScript IV (SSIV) kit (Thermo Fisher Scientific, USA) with random hexamer priming. Briefly, 11 μL of RNA was combined with 1 μL of 50 μM random hexamers and 1 μL of dNTP mix, incubated at 65 °C for 5 min, and snap-cooled on ice for 1 min. The reverse transcription mix was then completed by adding 4 μL SSIV buffer, 1 μL DTT, 1 μL RNase inhibitor, and 1 μL SSIV reverse transcriptase. Reactions were incubated at 25 °C for 5 min, 50 °C for 15 min, and heat inactivated at 80 °C for 10 min.

## PCR amplification, library preparation and barcoding

Complementary DNA (cDNA) was amplified using LongAmp Taq DNA Polymerase (New England Biolabs, USA) and influenza-specific primers[57]. Each 20 μL reaction contained 4 μL nuclease-free water, 1 μL primer mix (100 μM), 10 μL LongAmp Taq mix, and 5 μL cDNA. PCR cycling was performed as follows: 94 °C for 4 min; 35 cycles of 94 °C for 30 s, 50 °C for 30 s, and 72 °C for 1 min 30 s; followed by a final extension at 65 °C for 10 min and a 4 °C hold. Amplicons were quantified with a Qubit High Sensitivity dsDNA assay (Thermofisher, USA) following the manufacturer's instructions and diluted to 100 ng in 9 μL nuclease-free water. Rapid barcoding was performed using the Oxford Nanopore Rapid Barcoding Kit 24 V14 (SQK-RBK114.24) by adding 1 μL of barcode (RB01–RB24) to each sample. Reactions were incubated at 30 °C for 2 min and 80 °C for 2 min, then snap-cooled on ice. Barcoded samples were pooled and purified using a 1:1 ratio of AMPure XP beads (Beckman Coulter, USA) and DNA eluted in 15 μL Elution Buffer. Final DNA was quantified with a Qubit fluorometer.

## Nanopore sequencing, genome assembly and analysis

Sequencing was performed using Oxford Nanopore's GridION system. First, flow cell priming and loading were performed according to the manufacturer's protocol. A mixture of 30 μL Flow Cell Tether and 1170 μL Flow Cell Flush was applied through the priming port, followed by incubation for 5 min. During this time, 1.5 μL Rapid Adapter was combined with 3.5 μL adapter buffer, and 1 μL of this mix was added to 11 μL of the purified DNA library and incubated at room temperature for 5 min. The flow cell was re-primed with an additional 200 μL of the Flush/Tether mix. The sequencing mix was prepared as follows: 37.5 μL Sequencing Buffer, 25.5 μL Library Beads, and 12 μL DNA library, and the mix was loaded onto the flow cell and sequenced.

Following sequencing on the GridION platform, raw fastq files were processed using nf-flu workflow (CFIA−NCFAD/nf-flu, version 3.2.0; Kruczkiewicz et al., 2023; DOI: 10.5281/zenodo.8068086). The workflow, implemented in Nextflow, performs quality control, adapter trimming, host read removal, reference-based genome assembly, consensus sequence generation, variant calling and subtype reports for influenza A and B viruses. Analyses were run using the nanopore configuration and default parameters specified. Consensus sequences (nucleotide and amino acid) were aligned per segment with the reference genome using MAFFT[58] in Geneious Prime® 2025.1.2.

## Statistical analysis

Statistics were performed and significance was calculated as indicated where appropriate using GraphPad Prism 10 Software. $p$ values less than 0.05 were considered significant.

## Ethics statement

All animal experiments were conducted after obtaining prior approval from the Institutional Animal Care and Use Committee of Rocky Mountain Laboratories, National Institutes of Health(Protocol 2024-029E). Experiments were carried out in an Association for Assessment

and Accreditation of Laboratory Animal Care International-accredited facility, following the guidelines and basic principles in the Guide for the Care and Use of Laboratory Animals, the Animal Welfare Act, US Department of Agriculture, and the US Public Health Service Policy on Humane Care and Use of Laboratory Animals.

**Reporting summary**

Further information on research design is available in the Nature Portfolio Reporting Summary linked to this article.

## Data availability

All data supporting the findings of this study have been deposited on Figshare and are publicly available. https://doi.org/10.6084/m9.figshare.30574997. Source data are provided with this paper.

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

## Acknowledgements

This work was supported by the Intramural Research Program of the National Institute of Allergy and Infectious Diseases (NIAID), National Institutes of Health (NIH). The contributions of the NIH authors are considered Works of the United States Government. The findings and conclusions presented in this paper are those of the authors and do not necessarily reflect the views of the NIH or the U.S. Department of Health and Human Services. We thank Dr. Todd Davis at the Centers for Disease Control and Prevention (CDC), Decatur, Georgia, USA, for providing the Human HPAIV H5N1 isolate A/Texas/37/2024 (EPI_ISL_19027114). We also thank Dr. Richard Webby at St. Jude Children's Research Hospital, Memphis, Tennessee, USA, and Dr. Andrew Bowman at The Ohio State University, Columbus, Ohio, USA, for providing the Bovine HPAIV H5N1 isolate A/bovine/Ohio/B24OSU-342/2024 (EPI_ISL_19178076). We thank Erika Schwarz, Aracely Ospina Lopez, Nathaniel Antonioli, and Enrico Di Castro Young of the Molecular Diagnostics Section at the Montana Veterinary Diagnostic Laboratory, as well as Jennifer Ramsay and Matthew Becker of the Montana Fish, Wildlife & Parks Wildlife Health Laboratory, for providing the mountain lion specimen. This enabled tissue collection and isolation of the Mountain Lion HPAIV H5N1 isolate A/mountain lion/Montana/1/2024 (EPI_ISL_19083124). We are grateful to the Office of the Chief, RML, NIAID, NIH for their support within the high containment facility and the animal care staff of the Rocky Mountain Veterinary Branch, NIAID, NIH for their assistance during the study. We thank Ryan Kissinger, Visual and Medical Arts Section, RML, NIAID, NIH for preparing the graphical illustration (Fig. 1A). We acknowledge Dr. Joydeep Nag, Florida Research and Innovation Centre, Cleveland Clinic Lerner Research Institute for providing critical comments and valuable suggestions during the preparation of this manuscript.

## Author contributions

Conceptualization: R.K.M., V.J.M. Methodology: R.K.M., C.K.Y., V.J.M. Investigation: R.K.M., F.K.K., J.E.S., S.G., J.P.S., A.W., K.C., B.J.S., C.C., C.S., G.S., Ed.W., Nv.D., C.K.Y., V.J.M. Visualization: R.K.M., G.S., V.J.M. Supervision: V.J.M.

## Competing interests

The authors declare no competing interests.
