## [Transparent Peer Review file · Nature Communications]

Increased contact transmission of contemporary Human H5N1 compared to Bovine and Mountain Lion H5N1 in a hamster model

Corresponding Author: Dr Vincent Munster

Version 0:

Reviewer comments:

Reviewer #1

(Remarks to the Author)

The study by Reshma Koolaparambil Mukesh and colleagues investigates the pathogenicity and transmissibility of clade 2.3.4.4b H5N1 viruses derived from human, bovine, and mountain lion hosts using the Syrian hamster model. The experimental design is sound, and the study presents data relevant to understanding host-specific differences in H5N1 virus behavior.

However, in its current form, the manuscript does not offer significant new insights beyond what has already been published. Notably, a recent study by Iwatsuki-Horimoto et al. (2024), which the authors have cited, used the same HPAI H5N1 virus isolate, A/Texas/37/2024 (TX/37), in six-week-old Syrian hamsters. That study already demonstrated that Syrian hamsters could serve as a viable model for H5N1 infection, and it established key findings regarding the direct contact transmission of TX/37 and the absence of airborne transmission. These major conclusions are largely reiterated in the current manuscript.

The primary novel aspect of this study is the comparison between human, bovine, and mountain lion isolates, and the potential influence of the PB2-E627K mutation on pathogenicity and transmissibility. However, the authors do not sufficiently explore this angle. For instance, while the mutation is mentioned, its functional relevance is not thoroughly examined in the context of the observed outcomes. A deeper analysis of how this and other mammalian adaptation markers may contribute to inter-isolate differences would strengthen the manuscript considerably.

To enhance the value of this study, the authors are encouraged to address the following points:

Explicitly differentiate the findings of this study from those of Iwatsuki-Horimoto et al. (2024). What new insights are gained from including the bovine and mountain lion isolates, and how do these findings extend our understanding of zoonotic potential?

The manuscript (line 288) links increased viral shedding of the human isolate to its enhanced transmissibility. However, as shown in Figure 2, viral replication levels between the human and bovine isolates appear comparable. Could the authors discuss alternative mechanisms, such as differing infectious doses, immune responses, or the presence of PB2-E627K, that might explain the disparity in transmission?

The role of adaptation markers such as PB2-E627K should be more comprehensively addressed. How might these markers influence viral replication or transmissibility in the hamster model? Are there any other notable mutations distinguishing the isolates?

The discussion could benefit from a more detailed comparison with recent ferret studies (e.g., Pulit-Penalosa et al., Einfeld et al.). These studies used varying transmission protocols (direct vs. airborne; with or without fomites), which may impact the interpretation of transmission dynamics. A brief commentary on how these methodological differences influence comparisons with the hamster data would provide useful context.

The authors should clarify whether they observed signs of systemic infection (e.g., viremia, extrapulmonary dissemination) in the hamster model. Since systemic involvement has been reported in ferrets infected with similar isolates, this comparison

would help define the model's relevance and limitations.

Since the authors conclude that the hamster model complements established models, it would strengthen the manuscript to discuss the comparative advantages and limitations of hamsters versus ferrets, particularly in terms of pathogenicity, transmission dynamics, and practicality.

Reviewer #2

(Remarks to the Author)

The authors compared the pathogenicity and transmissibility of three recent H5N1 influenza virus isolates—obtained from bovine, mountain lion, and human hosts—in a hamster model. Changes in body weight and survival rates of infected hamsters were similar among the three isolates. Virus shedding in oropharyngeal swabs and virus titers in the nasal turbinates and lungs were also comparable, except for lower viral RNA levels in the lungs of hamsters infected with the mountain lion isolate. Histopathological analysis revealed milder lesions in the respiratory tracts of hamsters infected with the mountain lion isolate than in those infected with the human or bovine isolates. Although viral RNA levels in oropharyngeal swabs were similar across isolates, the human isolate showed slightly higher transmission efficiency than the other two.

Overall, the data appear sound for this type of study; however, several key elements are missing, as outlined below:

1. Because H5N1 viruses can replicate systemically, the authors need to quantify viral titers and assign pathological scores for the brain, liver, kidney, and spleen of infected hamsters at four days post-infection.
2. The authors should sequence the viruses recovered from contact hamsters to confirm genetic changes during transmission.

Version 1:

Reviewer comments:

Reviewer #2

(Remarks to the Author)

All of my concerns are correctly addressed in the revised manuscript.

Thank you very much for carefully reviewing our manuscript, "Increased contact transmission of contemporary Human H5N1 compared to Bovine and Mountain Lion H5N1 in a hamster model". We thank the reviewers for their time and effort and believe their comments have improved and increased the scientific impact of our study.

Reviewer

1:

The study by Reshma Koolaparambil Mukesh and colleagues investigates the pathogenicity and transmissibility of clade 2.3.4.4b H5N1 viruses derived from human, bovine, and mountain lion hosts using the Syrian hamster model. The experimental design is sound, and the study presents data relevant to understanding host-specific differences in H5N1 virus behavior.

However, in its current form, the manuscript does not offer significant new insights beyond what has already been published. Notably, a recent study by Iwatsuki-Horimoto et al. (2024), which the authors have cited, used the same HPAI H5N1 virus isolate, A/Texas/37/2024 (TX/37), in six-week-old Syrian hamsters. That study already demonstrated that Syrian hamsters could serve as a viable model for H5N1 infection, and it established key findings regarding the direct contact transmission of TX/37 and the absence of airborne transmission. These major conclusions are largely reiterated in the current manuscript.

The primary novel aspect of this study is the comparison between human, bovine, and mountain lion isolates, and the potential influence of the PB2-E627K mutation on pathogenicity and transmissibility. However, the authors do not sufficiently explore this angle. For instance, while the mutation is mentioned, its functional relevance is not thoroughly examined in the context of the observed outcomes. A deeper analysis of how this and other mammalian adaptation markers may contribute to inter-isolate differences would strengthen the manuscript considerably.

To enhance the value of this study, the authors are encouraged to address the following points:

We appreciate Reviewer 1 for highlighting these points. All comments and suggestions have been carefully reviewed and thoroughly addressed in the revised manuscript.

Explicitly differentiate the findings of this study from those of Iwatsuki-Horimoto et al. (2024). What new insights are gained from including the bovine and mountain lion isolates, and how do these findings extend our understanding of zoonotic potential? The manuscript (line 288) links increased viral shedding of the human isolate to its enhanced transmissibility. However, as shown in Figure 2, viral replication levels between the human and bovine isolates appear comparable. Could the authors discuss alternative mechanisms, such as differing infectious doses, immune responses, or the presence of PB2-E627K, that might explain the disparity in transmission?

We thank the reviewer for this insightful comment. Our study significantly extends the findings of Iwatsuki-Horimoto et al. by incorporating bovine and mountain lion isolates, which were not evaluated in the earlier work. Their work Iwatsuki-Horimoto et al. mainly focused on straightforward contact and airborne transmission, with low animal numbers, without any further characterization of the virological parameters.

In contrast, the inclusion of non-human mammalian isolates in the present study provides comparative insights into cross-species transmission potential. Specifically, the bovine and mountain lion isolates allow for a broader assessment of host adaptation features beyond the human context. In addition, detailed host and virological parameters were analyzed in our study (not present in the Iwatsuki-Horimoto study), such as survival, complete shedding profiles, pathological data, organ data, and full histopathological analyses).

Interestingly, whereas our data largely aligns with the finding in the paper of Iwatsuki-Horimoto et al. There are subtle differences as well, in particular the efficiency of transmission in our study is lower. We think this is largely contributed by the long-term exposure in the Iwatsuki-Horimoto study, whereas our transmission set-up, validated during our work with SARS-CoV-2, is more stringent with only limited duration of exposure of 48 hours and then animals were single housed to prevent cross-contamination.

Whereas the shedding initially looks very similar between the Bovine and Human isolate (by RT-PCR), the actual infectious shedding of the human isolate is significantly higher (figure 2D), we think that this effect is double. On one hand more of the human isolate comes out, but the PB2 627 adaptation results in relatively faster replication and likely reduces the infectious dose and overcomes the initial innate immune response, whereas the Bovine isolate is still transmitted, but quenched by the innate response and does not result in productive infection.

Despite limited transmission between the isolates, Human isolate demonstrated increased contact transmission. We agree that, beyond increased viral shedding, multiple

factors may contribute to the enhanced transmissibility of the human isolate compared to the bovine isolate. These possibilities including host immune responses, virus stability, and the presence of adaptation markers such as PB2-E627K have been discussed in the revised manuscript (Lines 345-3353).

The role of adaptation markers such as PB2-E627K should be more comprehensively addressed. How might these markers influence viral replication or transmissibility in the hamster model? Are there any other notable mutations distinguishing the isolates?

We have provided an overview of the amino acid differences between the viruses in supplemental table 1, For the only the PB2 627 is a known marker for increased replication based on the literature (Inventory of molecular markers affecting biological characteristics of avian influenza A viruses, Suttie et al).

The PB2-E627K is known to enhance influenza A virus replication. This is supported by our in vivo results in hamsters showing more shedding with the Human isolate (Figure 2, panel D).

To extend the in vivo data to virus replication kinetics between the isolates, we now have included results from a replication kinetics experiment using hamster respiratory cells and MDCK cells to assess replication at 33°C and 37°C. The human isolate performed similarly at both temperatures, whereas the Bovine and Mountain Lion isolates showed slower replication at 33C. In addition, when comparing the isolates with each other, the Human isolate replicated faster then the other two viruses.

The results of this additional experiment have been incorporated into the manuscript (Figure 6), and the results and discussion sections have been updated accordingly to reflect these findings (Lines 247-267, 360-365).

Further studies are warranted to elucidate the functional significance of these mutations in the hamster model. These points have now been explicitly addressed in the Discussion as part of the study's limitations (Lines 383-390), and we appreciate the reviewer's comment, which helped us clarify this aspect further.

The discussion could benefit from a more detailed comparison with recent ferret studies (e.g., Pulit-Penalozza et al., Einfeld et al.). These studies used varying transmission protocols (direct vs. airborne; with or without fomites), which may impact the interpretation of transmission dynamics. A brief commentary on how these methodological differences influence comparisons with the hamster data would provide useful context.

We agree that differences in transmission protocols may impact the efficiency of transmission, and we appreciate the reviewer's suggestion to elaborate on this aspect. In

the revised manuscript, we have now included a comparative discussion highlighting key methodological differences between our study and recent ferret studies.

We also discuss species-specific and experimental factors such as airflow control, housing design, and respiratory physiology, which further contribute to variability across models. These points have been added to the Discussion section (Lines 333-337) to aid interpretation and facilitate cross-study comparisons.

The authors should clarify whether they observed signs of systemic infection (e.g., viremia, extrapulmonary dissemination) in the hamster model. Since systemic involvement has been reported in ferrets infected with similar isolates, this comparison would help define the model's relevance and limitations.

Since we did not collect extrapulmonary tissues in the original study, we performed an additional study to assess systemic involvement following infection. We used group sizes of 6 animals per virus to obtain a detailed assessment of the extrapulmonary replication potential of viruses.

All viruses displayed a limited degree systemic replication potential, with 3/6 Bovine, 1/6 Mountain Lion and 2/6 Human and the systemic detection corresponded with high viral loads in the lower respiratory tract. This likely suggests that vascular leakage seeded the extrapulmonary replication of the viruses.

The results of this investigation have been included in the revised manuscript (Figure 3) and are described in detail in the results section (Lines 148-168).

Since the authors conclude that the hamster model complements established models, it would strengthen the manuscript to discuss the comparative advantages and limitations of hamsters versus ferrets, particularly in terms of pathogenicity, transmission dynamics, and practicality.

We appreciate your feedback. The points mentioned have been thoroughly addressed in the discussion section (Lines 288-292, 344-346)

Reviewer 2:

The authors compared the pathogenicity and transmissibility of three recent H5N1 influenza virus isolates—obtained from bovine, mountain lion, and human hosts—in a hamster model. Changes in body weight and survival rates of infected hamsters were similar among the three isolates. Virus shedding in oropharyngeal swabs and virus titers in the nasal turbinates and lungs were also comparable, except for lower viral RNA levels in the lungs of hamsters infected with the mountain lion isolate. Histopathological analysis

revealed milder lesions in the respiratory tracts of hamsters infected with the mountain lion isolate than in those infected with the human or bovine isolates. Although viral RNA levels in oropharyngeal swabs were similar across isolates, the human isolate showed slightly higher transmission efficiency than the other two.

We appreciate the time and effort you dedicated to reviewing our work, and we believe that your feedback has significantly strengthened the manuscript.

Overall, the data appears sound for this type of study; however, several key elements are missing, as outlined below:

1. Because H5N1 viruses can replicate systemically, the authors need to quantify viral titers and assign pathological scores for the brain, liver, kidney, and spleen of infected hamsters at four days post-infection.

Since we did not collect extrapulmonary tissues in the original study, we performed an additional study to assess systemic involvement following infection. We used group sizes of 6 animals per virus to obtain a detailed assessment of the extrapulmonary replication potential of viruses.

All viruses displayed a limited degree systemic replication potential, with 3/6 Bovine, 1/6 Mountain Lion and 2/6 Human and the systemic detection corresponded with high viral loads in the lower respiratory tract. This likely suggests that vascular leakage seeded the extrapulmonary replication of the viruses.

The results of this investigation have been included in the revised manuscript (Figure 3) and are described in detail in the results section (Lines 148-168).

2. The authors should sequence the viruses recovered from contact hamsters to confirm genetic changes during transmission.

As requested, we sequenced the viral genome recovered from positive contact hamsters and found no genetic changes associated during transmission. These results have been incorporated in the results section (Lines 234-235) of revised manuscript and materials and methods section has been modified with the protocol used (Lines 504-545).